# *Shaolinia*: A Fossil Link between Conifers and Angiosperms

**DOI:** 10.3390/plants13152162

**Published:** 2024-08-05

**Authors:** Xin Wang, Li-Jun Chen

**Affiliations:** 1State Key Laboratory of Palaeobiology and Stratigraphy, Nanjing Institute of Geology and Palaeontology, CAS Center for Excellence in Life and Paleoenvironment, Chinese Academy of Sciences, Nanjing 210008, China; 2Shenzhen Key Laboratory for Orchid Conservation and Utilization, National Orchid Conservation Center of China and Orchid Conservation & Research Center of Shenzhen, Shenzhen 518114, China

**Keywords:** angiosperms, conifers, carpel, homology, bract, placenta, Cretaceous

## Abstract

The flowering plants (angiosperms) are the dominant and defining group of the Earth ecosystems today. However, from which group and by what way flowers, especially their gynoecia (the key characteristic organs of angiosperms), are derived have been key questions in botany, and have remained unanswered despite botanists’ efforts over centuries. Such an embarrassing situation can be attributed to the lack of plants with partially enclosed ovules, which are supposed fill a position between gymnosperms and angiosperms. Here, we report a fossil plant that has apparent coniferous vegetative and reproductive characters but has a single seed partially wrapped by the subtending bract. Such a morphology suggests that a carpel of some angiosperms is equivalent to a lateral appendage (a bract plus its axillary seed) of this fossil. Such a non-traditional interpretation of the homology of angiosperm carpels is compatible with various new progresses made in botany and is in line with Tomlinson’s recent hypothesis. Together with other fossil evidence reported recently, it appears that gynoecia in angiosperms are derived in multiple ways.

## 1. Introduction

Flowering plants (angiosperms) can be distinguished from gymnosperms by their enclosed ovules before pollination [1,2,3]. But how such a feature evolved has been a puzzle for botanists for a long time. Formerly, carpels in angiosperms were supposed to be derived from megasporophylls that bear ovules along their margins [4]. But so far there is no fossil evidence favoring this hypothesis. On the contrary, there is increasing evidence rejecting this hypothesis [5,6,7,8,9,10,11,12,13,14,15,16,17,18]. To answer this question, fossil evidence is the only reliable source for morphological information of plants in the past. As angiosperms and gymnosperms are distinguished from each other by the status of ovules before pollination, being enclosed or naked [3], seeking an ovule morphology intermediate between those two states (namely, semi-enclosed) in a fossil plant is apparently critical.

## 2. Materials and Methods

A new specimen, including two facing slabs, was collected from the Yixian Formation (Lower Cretaceous) of Erdaogou, Jianchang, Liaoning (119.7451° E, 40.5288° N). The specimen includes physically connected conifer-like vegetative and reproductive organs. It was photographed using a Nikon D800 digital camera, and details were observed and photographed under a Leica M205A stereomicroscope equipped with a Leica DFC450C digital camera. A seed was removed from the specimen and observed using a Nikon SMZ1500 stereomicroscope equipped with a Nikon DS-Fi1 digital camera and a Leo 1530 scanning electron microscope (SEM). All photographs were saved in TIFF format and organized for publication using Photoshop 7.0.

## 3. Results

***Shaolinia*** gen. nov.

**Generic diagnosis**: Branch rigid, straight, with helically arranged leaves. Leaf *Juniperus*-like, straight or slightly curving. Cone-like reproductive organs alternately arranged along the branch. Reproductive organ including more than 40 lateral appendages helically arranged around a central axis. Each lateral appendage comprising a single axillary seed and subtending bract wrapping the seed from the bottom and laterals. Bract with a pointed tip, gaping adaxially. Seed round-shaped, with isodiametric epidermal cells.

**Type species**: *Shaolinia intermedia* gen. et sp. nov.

**Etymology**: *Shaolinia* dedicated to Dr. Shaolin Zheng, a senior Chinese palaeobotanist.

**Stratigraphic horizon**: the Yixian Formation.

***Shaolinia**intermedia*** gen. et sp. nov.

**Specific diagnosis**: the same as the genus.

**Description**: The specimen includes two facing parts of the distal portion of a woody branch, 68 mm long, 17 mm wide, preserved as a partially coalified compression and impressions in yellowish tuffaceous siltstone (Figure 1A,B). The branch is about 1.1 mm wide in the proximal portion, tapering distally (Figure 1A,B). There are several small axillary lateral branches up to 4.9 mm long, with a few leaves (Figure 1A,B). The leaves are *Juniperus*-like, 0.9–4.4 mm long and 0.2–0.5 mm wide, straight or slightly curving to the distal of the branch (Figure 1A,B and Figure 2A). Four cone-like reproductive organs are inserted alternately along the branch (Figure 1A,B and Figure 2B–D). The cone-like organ is 7.4–10.9 mm in length and 4.2–4.7 mm in diameter, with over fifteen lateral appendages helically arranged around a central axis (Figure 1A,B and Figure 2B–E). Each lateral appendage is round-triangular in shape in adaxial view, about 2.3 mm long, 0.8 mm high, and 1.5 mm wide, including an axillary seed and a subtending bract, and the latter wraps the former from the bottom and laterals (Figure 2B–E and Figure 3A,B). The seed is round in shape, 0.63–0.78 mm × 0.81–1.06 mm, more or less flattened during the fossilization; the seed coat has isodiametric epidermal cells (Figure 2D–F, Figure 3A,B and Figure 4A–D). The bract has a pointed tip, folding longitudinally and adaxially, gaping adaxially (Figure 2D,E, Figure 3A,B and Figure 5A–C).

**Etymology**: *intermedia*, Latin for the morphology of the lateral appendages that fall between carpels in some angiosperms and lateral appendages of conifers.

**Holotype**: JCEDG0001.

**Depository**: National Orchid Conservation Center of China and Orchid Conservation & Research Center of Shenzhen, Shenzhen, China.

**Remarks**: We refrain from using the terms like “carpels” and “flowers” in our description as they would imply we are treating *Shaolinia* as an angiosperm, although we do not discount the possibility that subsequent evolution from *Shaolinia* may have produced a true angiosperm bearing flowers and carpels.

The term “bract” is equivalent to that in conifers, and the “axillary seed” is equivalent to and comparable to the ovuliferous scale, which is more reduced (reduced into a single seed) than in conifers.

It is noteworthy that some conifers from the Cretaceous, although belonging to different groups, demonstrate features more or less similar to *Shaolinia*. For example, *Austrohamia acanthobractea* is similar to *Shaolinia* in terms of lacking a scale and wings on both sides of the seed [19], but its two seeds per bract [19] is a feature distinct from one seed per bract in *Shaolinia*.

*Chengia laxispicata* is a reproductive organ connected to other vegetative organs from the Early Cretaceous, and is identified as an element of Gnetales [20]. Its reproductive organ is a lax spike and was interpreted as having micropylar tubes [20]. These features distinguish *Chengia laxispicata* from *Shaolinia*, which has stout reproductive organs and lacks a micropylar tube.

*Jianchangia verticillata* is another ephedroid plant reported from the Early Cretaceous [21]. As the title of the original paper suggests, the fossil plant has “unusual bract morphology” that has never been seen in any ephedroid plants, casting doubt over the identification of this fossil. This doubt is further strengthened by the lack of a clear figure of the so-called micropylar tube as well as a complete lack of any information on “one or two outer envelopes enclosing an inner ovule” [21]. Therefore, the taxonomic position of *Jianchangia verticillata* is tentative. However, the spiny appearance of the “ovulate cone” [21] of *Jianchangia verticillata* alone distinguishes the plant from *Shaolinia*.

*Ephedra carnosa* was interpreted as an ephedroid plant from the Early Cretaceous, as the fossil had micropylar tube characteristic of Gnetales (and other groups) [22]. However, this interpretation is shaky in regard to its “micropylar tube”. In their Figure 4E, Yang and Wang said that there was “a straight micropylar tube”. However, it seems clear that there are two (not one) separated linear parts. It is difficult to call these two parts ONE micropylar tube, since as far as it is known, there is only one micropylar tube in a single *Ephedra* seed. Instead, similar structures are identified as a seed with a micropyle beak inside a fruit with a style [13]. This alternative interpretation is favored by the faint expanding base of the upper “micropylar tube” in Figure 4E of Yang and Wang [22]. The lack of such “micropylar tubes” in *Shaolinia* distinguishes *Shaolinia* from *Ephedra carnosa*, which actually has its seed fully enclosed [13] (rather than partially enclosed, as in *Shaolinia*).

*Qingganninginfructus* is a Jurassic angiosperm reported from the northwest of China [23]. This fossil is distinct from *Shaolinia* in its full enclosure of an anatropus bitegmic seed [23]. The vegetative parts of *Qingganninginfructus* are still unknown.

Lateral position of micropyle in *Shaolinia* is seldom seen in conifers. This position may be related to its non-orthotropous orientation, just as seen in a Triassic conifer that demonstrates certain similarity to some angiosperms, *Combina triassica* [24], which appears to have a skewed basal micropyle. Another Palaeozoic conifer called *Ullmania* [25] has its micropyle oriented adaxially; however, its micropyle is centered in the ovule (not skewed to either side of the ovule).

## 4. Discussion

Angiosperms can be distinguished from their gymnosperm peers by their enclosed ovules (angio-ovuly) before pollination [1,2,3], a character that ensures pollination is carried out in an angiospermous (not gymnospermous) way (although some angiosperms, i.e., *Reseda* [26], never fully enclose their ovules/seeds). However, how such a feature is derived from former gymnosperms with naked ovules is a crucial question of interest in botanical studies on the origin of angiosperms, because the answer to this question will determine the relationship between angiosperms and gymnosperms and unite both into an integral system of seed plants. To answer this question, a fossil plant with a semi-enclosed ovule is of key importance. Although conifer-like in vegetative morphologies, *Shaolinia* has its seed wrapped by its subtending bract. Despite the seed/ovule not being fully enclosed, as in typical angiosperms, the seed-wrapping tendency of *Shaolinia* is intriguing because it is not hard to extrapolate further evolution from conifers and *Shaolinia* to the fully enclosed ovules/seeds of angiosperms. *Shaolinia* is not alone in terms of such an interpretation, as the recently reported *Combina triassica* has been interpreted in exactly the same way [24]. Such a congruous ovule-enclosing tendency in conifer-like *Shaolinia* and *Combina* makes it intriguing to relate conifers with angiosperms, an evolutionary scenario anticipated by some botanists [3].

Axillary branching is a pattern frequently seen in angiosperms and many gymnosperms. Its history can be dated at least back to the Middle Pennsylvanian [25,27], and is the Bau-plan underlying the lateral appendages of conifer cones. As proven by Florin’s works [28,29,30,31], the lateral appendages in the reproductive organs of Cordaitales and Coniferales are compound organs comprising an axillary ovule-bearing branch and a subtending bract. Various metamorphisms of these two parts have given rise to diverse cones/gynoecia in these two groups. This information has been well known for a long time. However, its implication for the homology of carpels has rarely been explored before, most likely due to the dominance of Arber and Parkin’s hypothesis [4], in which a carpel was assumed to have been derived from a so-called “megasporophyll”, which, however, has been proven non-existent [9,10]. Most botanists have been misled by such a groundless speculation. The first obvious light elucidating the homology of carpels emerged more than 20 years ago when Roe et al. [5] demonstrated that the ovules in *Arabidopsis* are independent of the carpel wall and borne on a branch. Morphologically, the most significant progress was made recently when Zhang et al. [6] revealed that the carpels in *Michelia figo* (Magnoliaceae) are composite organs derived from a former axillary ovule-bearing branch and its subtending foliar part. The varying morphology of carpels in a single tree of *Michelia figo* [6] demonstrates a great resemblance to the lateral appendages of *Shaolinia* in term of axillary seeds and subtending foliar parts. Such observations make it obvious that the axillary branching frequently seen in many seed plants is a feature shared by conifers (besides others) and angiosperms, and a carpel may well be derived from an axillary ovule-bearing branch (equivalent to a placenta) and its subtending foliar part (equivalent to the ovarian wall). After considering all potential alternatives, Doyle thinks that “the carpel could represent a leaf and a cupule-bearing axillary branch” [32]. This is in line with the view of developmental geneticists that separate the placenta from the carpel wall [32,33], which appears to have been adopted by plant morphologists [34]. This idea actually is not far different from the concept of a “gonophyll” as advanced by Melville [35]. Configuration and organization of lateral appendages in *Shaolinia* is comparable to the carpels in *Illicium lanceolatum* (Schisandraceae) [36], and the only difference between these two lies in the extent that the carpel wall (=bract) is closed. This speculation has been favored by a Triassic conifer-like reproductive organ, *Combina triassica* [24]. It is noteworthy that *Eoantha* in Figure 3c of Krassilov [37] is comparable to the lateral appendage of *Shaolinia* in term of an ovule/seed wrapped by a foliar part from the abaxial and laterals. Similarly, although the ventral suture of Late Jurassic and Early Cretaceous *Dirhopalostachys rostrata* [38] is narrower than the adaxial gap of the bract in *Shaolinia*, their lateral appendages are of similar organization. Comparing *Combina*, *Dirhopalostachys*, *Shaolinia*, and *Illicium*, it is easy to figure out the underlying homology of the lateral appendages and carpels in these taxa. Now, the occurrence of *Shaolinia* seems to reinforce and strengthen the evidence chain for the derivation of carpels in angiosperms from bract-scale complexes in conifers, as suggested by Tomlinson [3], and solves the recalcitrant problem of the origin of angiosperms and their carpels. It is intriguing to note that some extant conifers, i.e., Araucariaceae, also demonstrate a certain tendency to enclose their seeds.

Based on a comparison of gynoecium organization in early angiosperms [12,13,14,15,16,17,18,39,40,41,42,43,44,45], it is obvious that the gynoecia of these early angiosperms are diversified in morphology and organization. Although gynoecia in Magnoliales and Amborellales cannot be excluded from the list of ancestral taxa, there are other types of gynoecia that may be plesiomorphic and derived independently throughout geological history. For example, the Early Jurassic *Nanjinganthus* is hard to accept for many because of its “noncarpellate” inferior ovary [12,43,46], a feature that used to be thought as much derived in angiosperm systematics. The inferior ovary in *Nanjinganthus* may be derived from a typical flower axis through distal expansion and invagination, a scenario formerly inconceivable although such a tendency has been seen in *Amborella*, the basalmost extant angiosperm recognized currently [47]. Previous works suggested that angiosperm characters may have evolved independently in various taxa [35,48,49,50]. This is also the opinion of botanists studying extant plants [37,51,52]. It appears more plausible that the gynoecia in angiosperms were derived from different taxa independently [33,34,35].

Interestingly, *Araucaria columnaris* appears to have its seed (at least partially) enclosed, thus the term “carpel” was applied in a description [53]. This feature is so intriguing that Krassilov and Barinova compared it with some angiosperms. However, it appears to be a premature treatment: although, as in some other conifers [1], seeds are enclosed in *Araucaria columnaris*, it does not mean that it is an angiosperm, as the timing of pollination and enclosure is pivotal in determining a plant being an angiosperm [1], and Krassilov and Barinova did not provide any evidence indicating that the ovule enclosure occurs before pollination in *Araucaria columnaris*. However, it should be borne in mind that these cases indicate that some conifers tend to protect their ovules/seeds, just as in angiosperms. All these data seem to confirm the universal existence of ODC (offspring development conditioning) in plant evolution [54].

## 5. Conclusions

The new fossil sheds more light on the evolution of female reproductive organs, suggesting potential homology between some conifers and angiosperms. In line with previous fossil reproductive morphology, there appear to be various independent ways to derive the reproductive organs (flowers) in angiosperms.

## Figures and Tables

**Figure 1 plants-13-02162-f001:**
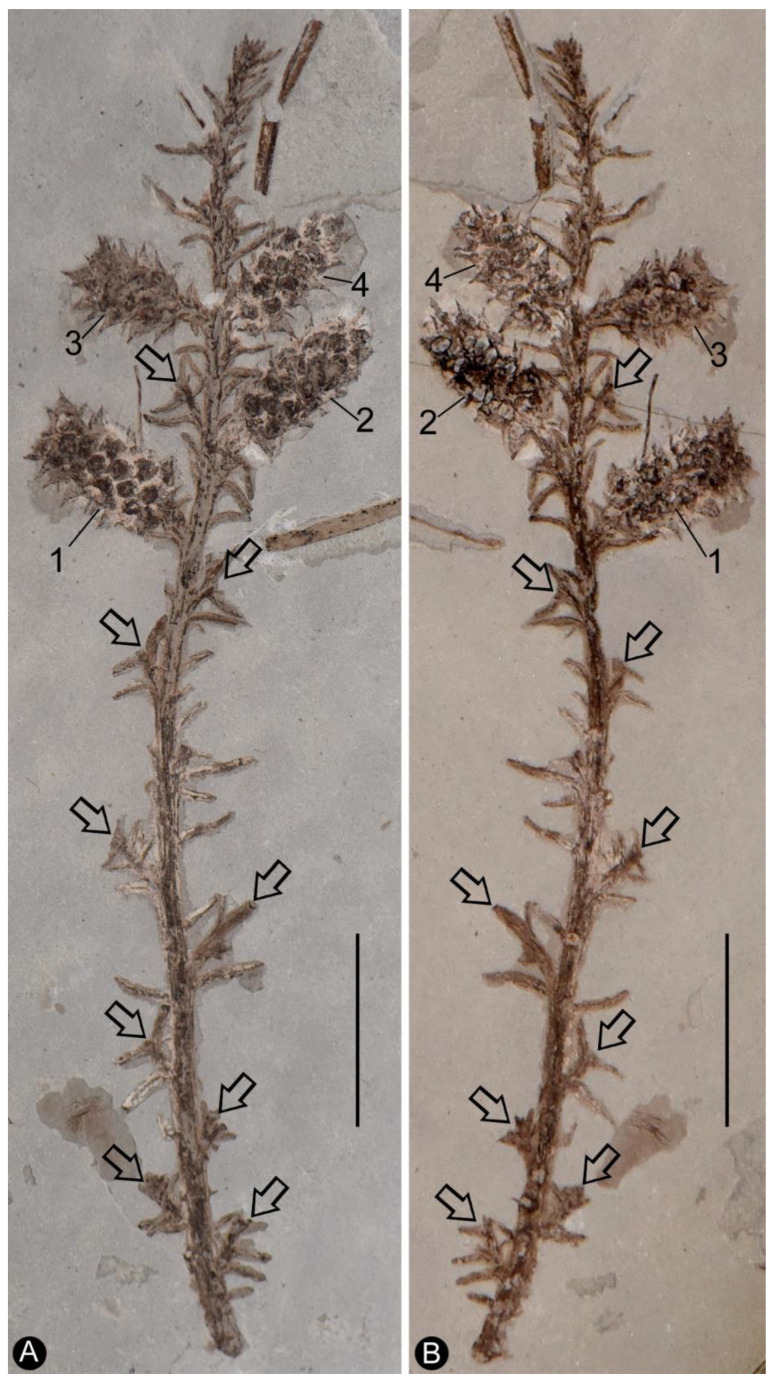
General view of *Shaolinia intermedia* gen. et sp. nov. (**A**,**B**) Two counterparts of the same specimen, showing several axillary branches (arrows), leaves and four cone-like reproductive organs (1–4) arranged along the branch. Scale bar = 1 cm.

**Figure 2 plants-13-02162-f002:**
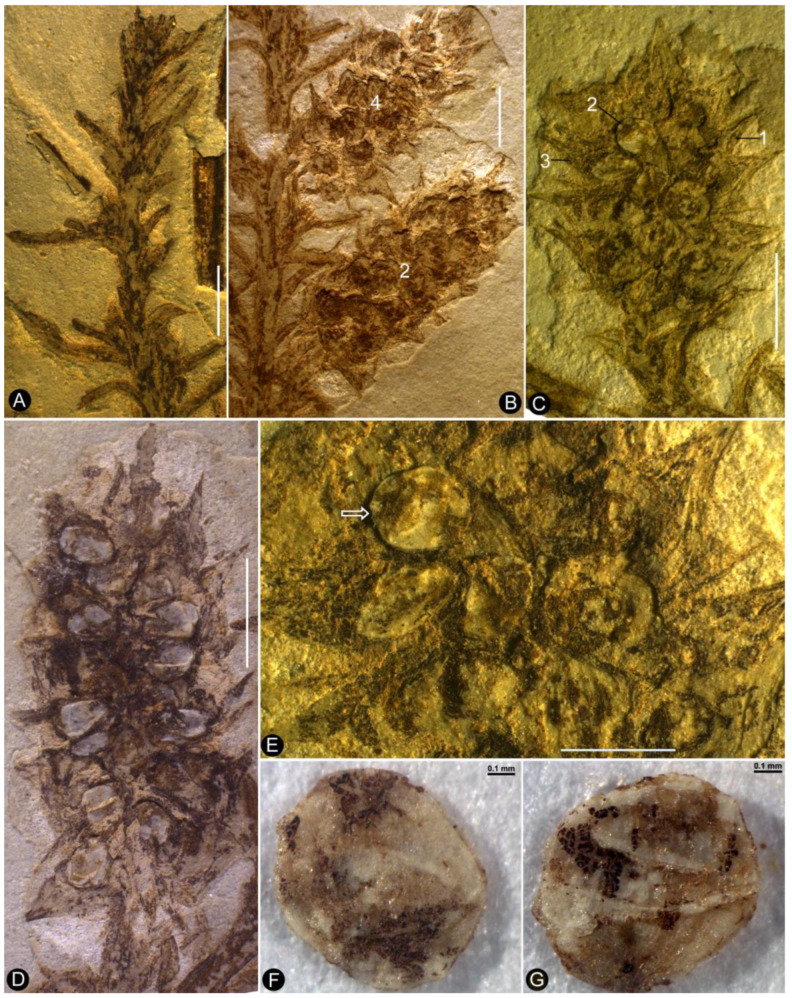
Leaves and reproductive organs of *Shaolinia intermedia*. (**A**) The distal portion of the branch with helically arranged leaves. Scale bar = 2 mm. (**B**) Detailed view of the reproductive organs (Nos. 2 and 4). Note the helically arrange lateral appendages. Scale bar = 2 mm. (**C**) Detailed view of reproductive organ No. 3, with helically arranged lateral appendages. Scale bar = 2 mm. (**D**) Detailed view of reproductive organ No. 2, showing seeds inside the lateral appendages. Scale bar = 2 mm. (**E**) Detailed view of the distal portion of the reproductive organ shown in (**C**). The arrowed seed is removed and shown in (**F**,**G**) and Figure 4A–D. Scale bar = 1 mm. (**F**,**G**) Detailed views of the seed removed from the organ shown in (**E**). Scale bar = 0.1 mm.

**Figure 3 plants-13-02162-f003:**
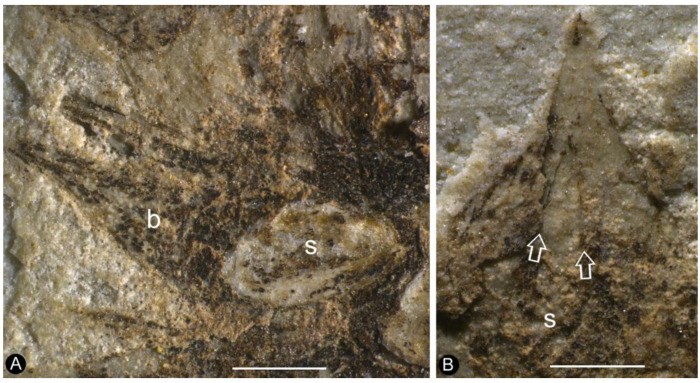
Lateral appendages with in situ seed inside and an adaxial gap. (**A**) Detailed side view of lateral appendage No. 3 in Figure 2C, showing the seed (s) wrapped by the bract (b). Scale bar = 0.5 mm. (**B**) Detailed side view of top lateral appendage in Figure 2C, showing the adaxial gap with two margins (arrows) of the bract. Scale bar = 0.5 mm.

**Figure 4 plants-13-02162-f004:**
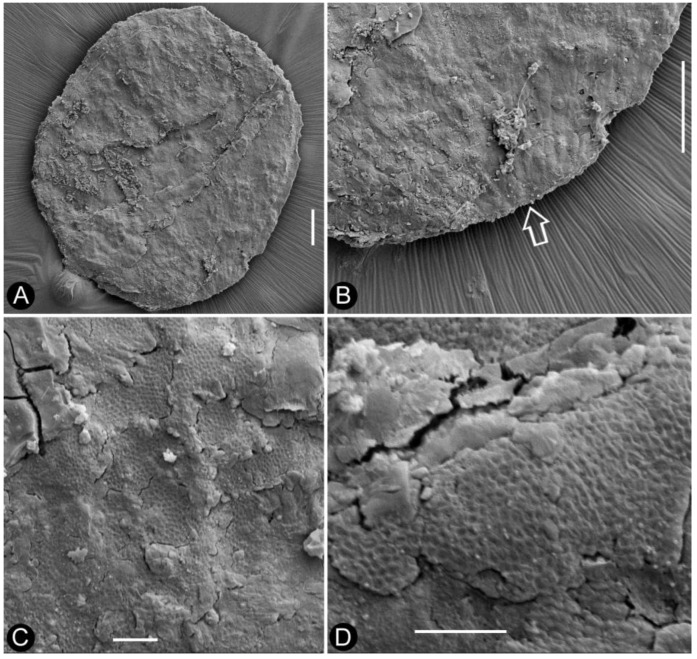
SEM view of the *in situ* seed found in *Shaolinia intermedia*, from the reproductive organ No. 2 shown in Figure 2C,E–G. (**A**) Seed in whole. Scale bar = 0.1 mm. (**B**) Converging cellular arrangement suggestive of possible micropyle (arrow) of the seed. Scale bar = 0.1 mm. (**C**) Seed coat sculpture. Scale bar = 10 μm. (**D**) Detailed cellular view of the seed coat sculpture. Scale bar = 10 μm.

**Figure 5 plants-13-02162-f005:**
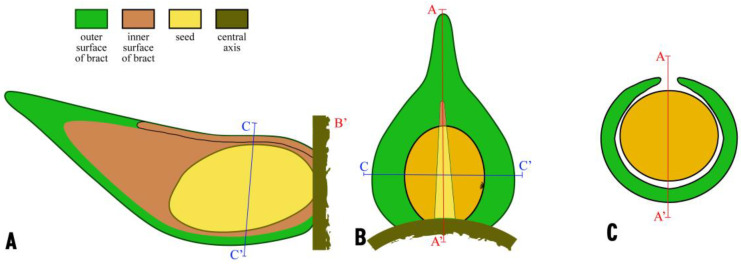
Sketches of the lateral appendages of *Shaolinia intermedia* showing spatial relationship between subtending bract and axillary seed. Not to scale. (**A**) Longitudinal radial profile of a lateral appendage showing a bract and an adaxial seed attached to the central axis and partially wrapped. The position of (**C**) is marked. (**B**) Adaxial view of a lateral appendage showing a bract and a partially wrapped adaxial seed inside. Note the adaxial gap of the bract and micropyle on the right side of the ovule. The positions of (**A**,**C**) are marked. (**C**) Cross view of a lateral appendage showing a bract and an axillary partially wrapped adaxial seed. The position of (**A**) is marked.

## Data Availability

All data generated or analyzed during this study are included in this published article.

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
