# Peer review of "Shaolinia: A Fossil Link between Conifers and Angiosperms"

_plants, 2024, doi:10.3390/plants13152162_

Round 1
Reviewer 1 Report
Comments and Suggestions for Authors
First thing first – the title – “Shaolinia: A Fossil Post Between Conifers and Angiosperms”. I don’t understand what the authors mean here with the word ‘post’. I looked it up in the dictionaries and the closest meaning that makes sense to me is “noun (2) 4 obsolete : courier” in Meriam Wester Dictionary (https://www.merriam-webster.com/dictionary/post). Do the authors mean that this fossil serves as a “messenger” between the conifers and the angiosperms? If so, please note that this usage (meaning courier) is obsolete. An obsolete usage means that a word that is no longer in use or no longer useful. If not, I am unable to figure out the meaning of post in the title. One of the goals of having your paper published is to communicate your information (or your discovery) with your colleagues or better yet, the general public. If your audience have a difficult time to understand your title, that is not a good start communicating your information. I am unable to find the meaning of the word post in both the American Heritage Dictionary and the Oxford English Dictionary that makes sense being used here in this title. If the authors’ intention is to use the word post to mean “link” or “transitional form” in the title, then use one of the two.
In its current state, the level of English throughout this manuscript does not meet the journal’s requirements. The English text (grammar, wrong choice of words, structure of sentences, odd phrasing, etc.) needs to be improved. I think that it is the authors' responsibility to proof their manuscript for English before submission. I recommend that the authors seek the assistance of a native English speaker or an expert linguist, preferably one who has some knowledge of geology and/or palaebotany, to carefully go through the revised manuscript to improve the English grammar, choice of words and the punctuation. The authors may also consult a professional editing service.
As a non-native English speaker myself, I am sympathetic to the challenges faced by non-English speakers. However, the English issue in this manuscript needs to be fully addressed prior to considering its publication in Plants. To illustrate my point, I have marked some suggested changes (note: not exhaustive) in the pdf file of the manuscript.
Interpretation of the fossil specimens
In figure 1, the authors interpret that there are several (eight to be specific, as indicated by the arrows) axillary branches. Why are they axillary branches, not the basal portions of the broken/detached axes of the cone-like reproductive organs? Please explain.
In generic diagnosis in section 3, Results, the first sentence ‘Plant woody, including branches, leaves, and connected cone-like organs.’ Please note that these are not ‘characters’ except ‘plant woody’ – even this ‘character’ is the authors assumption, not an observed ‘character’. Therefore, it is more appropriate to put this in the discussion, or perhaps in description, but not in generic diagnosis.
Please define the meaning of a “lateral appendage”. In diagnosis, the author state that “Each lateral appendage comprising a single axillary seed and sub-tending bract wrapping the seed from the bottom and laterals.” However, in figure caption (Fig. 2D), the authors state that “D. Detailed view of reproductive organ No. 2, showing seeds inside the lateral appendages.” Is a seed part of a lateral appendage? Or a seed and a lateral appendage together form a reproductive organ, and the reproductive organs form a cone-like structure? Then what about a bract? In Fig. 5, the authors do seem to indicate that a lateral appendage contains a bract and a seed by stating that “… a lateral appendage showing a bract and an adaxial seed attached…”.
With compressed specimens, it is extremely difficult to prove that these seeds are wrapped by the bracts. For example, in Fig. 3A, it seems to me that the seed (s) is fully exposed, i.e., not enclosed by the bract. Of course, it is likely that the seed is indeed enclosed (or partially enclosed) by the bract but the carbon material fell off, exposing the seed. But based on the image provided, I don’t see any clear evidence that the seed is enclosed (even partially) by the bract.
The authors’ observation is that the fossil specimen has “… a single seed partially wrapped by the subtending bract…”. Therefore, they suggest that this fossil represents “… a post (a transitional form or a missing link – my understanding) between gymnosperms and angiosperms…”. Then what about Nanjinganthus, “… a bona fide angiosperm from the Jurassic”? (see Fu et al., 2017; https://doi.org/10.1101/240226). And Qingganninginfructus? (see Han et al. 2023; https://doi.org/10.3390/life13030819). Discussions on the relationships between the new fossil from the Cretaceous and those angiosperms from the Jurassic would be very elucidating to the readers.
Again. How about fossils from other groups of plants that have partially enclosed ovules? For example, Ephedra carnosa (Yang and Wang, 2013; doi:10.1371/journal.pone.0053652) and Chengia laxispicata (Yang et al., 2013; http://www.biomedcentral.com/1471-2148/13/72), and others (see Yang et al., 2020; https://doi.org/10.1186/s12862-019-1569-y). Discussions on the relationships of the new fossil with these fossils would help the readers understand the evolution of enclosed ovules of angiosperms.
Of course, there are other conifers that seem to display partially enclosed seeds/ovules as well (see Dong et al., 2018, Figs. 10, 11; DOI: 10.1086/699665). Can we consider these fossils as transitional forms between conifers and angiosperms?
Final question: is this fossil a conifer or an angiosperm?

Reviewer 2 Report
Comments and Suggestions for Authors
The paper describes a new fossil that, according to the views of the authors provides information useful for understanding relationships between gymnosperms and angiosperms.
I have several comments and suggestions after reading the manuscript.
The problem is that angiosperms differ from gymnosperms in a combination of characters, not in single individual characters. Even some characters that are indeed important, can nevertheless vary within each group. The character of ovules enclosed before pollination is a good example. There are extant gymnosperms with ovules enclosed before pollination, such as Araucaria. There is extensive literature on female reproductive structures in Araucaria that should be cited and discussed here. Here is a relevant paper:
Krassilov, V., & Barinova, S. (2014). Carpel–fruit in a coniferous genus Araucaria and the enigma of angiosperm origin. Journal of Plant Sciences, 2(5), 159-166.
In fact, Valentin Krassilov published a lot of books and papers where he expressed ideas similar to those of the authors of the present manuscript. It is a pity that no work of Krassilov is cited here. I am listing below a few potentially useful references:
Krassilov, V. A. (1977). The origin of angiosperms. The Botanical Review, 43, 143-176.
Krassilov, V. A. (1991). The origin of angiosperms: new and old problems. Trends in Ecology & Evolution, 6(7), 215-220.
Krassilov, V. A. (1984). New paleobotanical data on origin and early evolution of angiospermy. Annals of the Missouri Botanical Garden, 577-592.
Krassilov, V. A. (2002). Character parallelism and reticulation in the origin of angiosperms. In Horizontal gene transfer (pp. 373-382). Academic Press.
Krassilov, V. A., & Bugdaeva, E. V. (1999). An angiosperm cradle community and new proangiosperm taxa. Acta Palaeobotanica, 1999, 111-127.
Krassilov, V. A. (1975). Dirhopalostachyaceae–a new family of proangiosperms and its bearing on the problem of angiosperm ancestry. Palaeontographica B, 153(1/3), 100-110.
Krassilov, V., & Barinova, S. (2014). «Flower» of Magnolia grandiflora is not flower and what about «basal angiosperms». Journal of Plant Sciences, 2(6), 282-293.
Krassilov, V. A., & Burago, V. I. (1981). New interpretation of Gaussia (Vojnovskyales). Review of Palaeobotany and Palynology, 32(2-3), 227-237.
Krassilov, V. A. (1997). Angiosperm origins: morphological and ecological aspects. Pensoft Publishers.
There are extant angiosperms in which ovules not fully enclosed before pollination, and these are not intermediate forms between angiosperms and gymnosperms, but just angiosperms. The most relevant reference is the following:
Endress, P. K., & Igersheim, A. (2000). Gynoecium structure and evolution in basal angiosperms. International Journal of Plant Sciences, 161(S6), S211-S223.
A good example of angiosperm gynoecium open at anthesis is Reseda. Here is one of relevant papers, but there are others:
ARBER, A. (1942). Studies in Flower Structure: VII. On the Gynaeceum of Reseda, with a Consideration of Paracarpy. Annals of Botany, 6(21), 43-48.
I suggest discussing the example of Reseda and adding references.
Another famous example of open gynoecium is Plathystemon (Papaveraceae). It is good to discuss it, too.
The authors state in the Introduction: “angio-sperms and gymnosperms are distinguished each other by the status of ovules before pollination, being enclosed or naked”. In fact, these are the two most typical conditions, but this cannot be used as the only technical criterium to distinguish between the two groups (especially for practical identification of a plant as either an angiosperm or a gymnosperm). Therefore, I suggest re-writing the Introduction by changing the accent. Also, while rejecting the theory that the carpels are megasporophylls, I suggest citing important publications by J.A.Doyle, at least the following one:
Doyle, J. A. (2008). Integrating molecular phylogenetic and paleobotanical evidence on origin of the flower. International Journal of Plant Sciences, 169(7), 816-843.
I suggest adding that the Yixian Formation belongs to the Cretaceous, if that is the case. The word Cretaceous could be then added to keywords.
“Seed round-shaped, with isodiametric sculpture”. – I do not understand what the authors mean under isodiametric sculpture.
I am not sure I fully understand Figure 5. It will be good to provide diagrams of sections in three perpendicular planes and to indicate the position of the micropyle. What is the narrow brown triangle above the seed in the right figure?
Having a convincing micropyle illustrated will be extremely important, because if the micropyle is not identified, then one can ask why the ovoid body is not a one-seeded fruit/carpel?
Round 2
Reviewer 1 Report
Comments and Suggestions for Authors
The cover letter states that Mike Pole has help with the English. But I doubt that Pole has actually read through the entire manuscript. Otherwise, the following examples of mis-usages should be caught by a native English speaker:
Etymology: Shaolinia dedicated to Dr. Shaolin Zheng, a senior the late Chinese palaeobotanist.
Description: The specimen includes two facing parts of the distal portion of a woody branch, 68 mm long, 17 mm wide, preserved as a partially coalified compression and im-pressions on in yellowish tuffaceous siltstone.
Use of hyphen instead of en dash. See Description. En dashes should be used in these places, not hyphens. These are perhaps the jobs of the style editor but I think that authors should be aware of this usage.
Fig. 3 Lateral appendages with in situ seed inside and an adaxial gap. A. Detailed side view of lateral appendage No. 3 in Fig. 2cC, showing the seed (s) wrapped by the bract (b). Scale bar = 0.5 mm. B. Detailed side view of top lateral appendage in Fig. 2cC, showing the adaxial gap with two margins (arrows) of the bract. Scale bar = 0.5 mm. --- Again, this may be of the job of the style editor, but the authors should be consistent with the usage.
Comments on the Quality of English Language
The cover letter states that Mike Pole has help with the English. But I doubt that Pole has actually read through the entire manuscript. Otherwise, the following examples of mis-usages should be caught by a native English speaker:
Etymology: Shaolinia dedicated to Dr. Shaolin Zheng, a senior the late Chinese palaeobotanist.
Description: The specimen includes two facing parts of the distal portion of a woody branch, 68 mm long, 17 mm wide, preserved as a partially coalified compression and im-pressions on in yellowish tuffaceous siltstone.
Use of hyphen instead of en dash. See Description. En dashes should be used in these places, not hyphens. These are perhaps the jobs of the style editor but I think that authors should be aware of this usage.
Fig. 3 Lateral appendages with in situ seed inside and an adaxial gap. A. Detailed side view of lateral appendage No. 3 in Fig. 2cC, showing the seed (s) wrapped by the bract (b). Scale bar = 0.5 mm. B. Detailed side view of top lateral appendage in Fig. 2cC, showing the adaxial gap with two margins (arrows) of the bract. Scale bar = 0.5 mm. --- Again, this may be of the job of the style editor, but the authors should be consistent with the usage.
Author Response
See the attached fille

Reviewer 2 Report
Comments and Suggestions for Authors
There are the following two phrases in the abstract: (1) This morphology suggests that a carpel in angiosperms is equivalent to a lateral appendage (a bract plus its axillary seed) of this fossil; (2) Together with other fossil evidence reported recently, it appears that gynoecia in angiosperms are derived in multiple ways.
There is a contradiction between these phrases. If the second phrase was right, then it is necessary to modify the first phrase as follows: This morphology suggests that a carpel OF SOME angiosperms is equivalent to a lateral appendage (a bract plus its axillary seed) of this fossil
Introduction. Angiosperms differ from gymnosperms in a combination of characters, not in a single character.
I still have problems with understanding Figure 5B. If that is an adaxial surface view, then I do not understand why do we see here the adaxial surface. The adaxial surface should be inside, facing the seed, and not visible from the outside. It will be good to add a third image with a section in the third (vertical tangential) plane (it will correspond to a vertical line in A and to a horizontal line in B). Also, the seed in B appears to be longer than in A.
The putative micropyle out of the median plane of the bract is intriguing. It will be important to add comparisons with other conifers (extant and extinct). Do you know other examples of a lateral micropyle? Please, specify the type of the ovule you assume for this fossil.
Please, change Dirhopalostachya rostrata to Dirhopalostachys rostrata (page 7, two times).
Page 8: “It appears more plausible that the gynoecia in angiosperms were derived from different taxa independently (33-35), an evolutionary scenario previously unexpected.” – I suggest removing the second part (after the comma), because just above you discuss such scenarios.
Page 8, the last paragraph before the Conclusions. As noted by Krassilov in his another publication, pollen of Araucaria does not germinate at the ovule micropyle. Please, try to follow this link: https://books.google.co.il/books?hl=ru&lr=&id=lXzlpRw_y38C&oi=fnd&pg=PA1&dq=krassilov+pensoft&ots=OxBkbqt8Wj&sig=ji1VTZZaGRXGJkWl4gz_zuo1C8U&redir_esc=y#v=onepage&q&f=false. He provides a reference to Fiordi 1984.
